# Genome-Wide Identification and Functional Evolution of NLR Gene Family in *Capsicum annuum*

**DOI:** 10.3390/cimb47100867

**Published:** 2025-10-21

**Authors:** Chong Feng, Qi Chen, Wenhao Liu, Tengfei Li, Tuo Ji

**Affiliations:** 1School of Life Sciences and Biotechnology, Shanghai Jiao Tong University, Shanghai 200240, China; fcsjtu@sjtu.edu.cn; 2College of Horticulture Science and Engineering, Shandong Agricultural University, Taian 271018, China; chihuoxiaochen@163.com (Q.C.);; 3Dezhou Academy of Agricultural Sciences, Dezhou 253012, China

**Keywords:** pepper, Nucleotide-binding leucine-rich repeat proteins, *Phytophthora capsici*, tandem duplication

## Abstract

*Capsicum annuum* (pepper) is a globally significant Solanaceous crop vulnerable to devastating pathogens such as *Phytophthora capsici*. Nucleotide-binding leucine-rich repeat (NLRs) proteins are crucial intracellular immune receptors mediating effector-triggered immunity (ETI). This study presents the comprehensive genome-wide identification and analysis of the NLR gene family in pepper using the high-quality ‘Zhangshugang’ reference genome. We identified 288 high-confidence canonical NLR genes. Chromosomal distribution analysis showed significant clustering, particularly near telomeric regions, with Chr09 harboring the highest density (63 NLRs). Evolutionary analysis demonstrated that tandem duplication is the primary driver of NLR family expansion, accounting for 18.4% of NLR genes (53/288), predominantly on Chr08 and Chr09. Analysis of promoter cis-regulatory elements (CREs) revealed enrichment in defense-related motifs, with 82.6% of promoters (238 genes) containing binding sites for salicylic acid (SA) and/or jasmonic acid (JA) signaling. Transcriptome profiling of *Phytophthora capsici*-infected resistant (*C. annuum* cv. CM334) and susceptible (*C. annuum* cv. NMCA10399) cultivars identified 44 significantly differentially expressed NLR genes, and protein–protein interaction (PPI) network analysis predicted key interactions among them, with Caz01g22900 and Caz09g03820 as potential hubs. This study elucidates the tandem-duplication-driven expansion, domain-specific functional implications, and expression dynamics of the pepper NLR family. It identifies conserved and lineage-specific candidate NLR genes, including *Caz03g40070*, *Caz09g03770*, *Caz10g20900*, and *Caz10g21150*. These findings provide valuable candidate gene targets for the development of molecular markers for pepper resistance to *Phytophthora capsici*.

## 1. Introduction

During long-term evolution, plants have developed sophisticated innate immune systems constituting the core defense against pathogenic microorganisms. This system primarily comprises two layers of defense mechanisms: Pattern-triggered immunity (PTI), activated by cell surface pattern recognition receptors (PRRs) sensing pathogen-associated molecular patterns (PAMPs) or host-derived damage patterns (DAMPs), and effector-triggered immunity (ETI), triggered by intracellular resistance (R) proteins recognizing specific pathogen effectors. The latter typically induces a stronger and faster defense response, including programmed cell death (hypersensitive response, HR), which effectively restricts pathogen colonization and proliferation [1,2]. NLR proteins serve as central executors of ETI. NLR proteins feature a characteristic modular structure: an N-terminal signaling domain (e.g., Toll/Interleukin-1 Receptor homology (TIR), Coiled-Coil (CC), or RPW8-like domain), a central conserved nucleotide-binding domain (NBS, Nucleotide-Binding Site), and a C-terminal leucine-rich repeat (LRR) domain responsible for effector recognition or protein interactions. This architecture enables NLRs to act as molecular switches, detecting direct or indirect effector interference via conformational changes and subsequently activating downstream immune signaling pathways [3,4]. Thus, NLR genes represent critical genetic resources for plant disease resistance and are pivotal subjects for evolutionary studies.

To counter rapidly evolving pathogens, the plant NLR gene family exhibits significant polymorphism and dynamics. Its expansion primarily relies on three mechanisms: tandem duplication, segmental duplication, and retrotransposition [5]. For instance, numerous NLRs in rice cluster near chromosomal telomeres, facilitating rapid generation of new resistance alleles through local amplification [6,7]. However, this “arms race” complicates NLR annotation, as highly variable LRR domains often generate pseudogenes, and truncated NLRs may confound functional predictions [8]. The NLR family is ubiquitous and forms large, complex entities in plant genomes, displaying unique evolutionary traits. Sequence variation, particularly in the hypervariable LRR domain, enables continuous adaptation to evolving pathogen effectors. Moreover, NLR gene numbers vary drastically between species and even among ecotypes (e.g., *Arabidopsis thaliana* contains ~150 NLRs, while *Oryza sativa* harbors ~500), driven primarily by gene duplication events, with tandem duplication being a major driving force in local cluster formation and family expansion [5]. Concurrently, the persistent “arms race” with pathogens subjects NLR genes to intense positive selection, resulting in rapid coevolution and neo-functionalization [9,10]. These characteristics pose significant research challenges: vast sequence diversity complicates genome annotation; functional validation of numerous redundant genes is labor-intensive; and the molecular mechanisms underlying their dynamic evolution (e.g., duplication, neo-/sub-functionalization, pseudogenization, adaptive selection) remain poorly understood [11,12]. Comprehensive characterization of the NLR family’s composition, evolutionary patterns, and functional diversification in specific species is essential to overcoming these challenges and harnessing valuable disease resistance resources.

Pepper (*Capsicum annuum*), a globally significant Solanaceous crop, provides essential vitamins, pigments, and flavor compounds, with an annual yield about 40 million tons (FAO, 2023). However, its production is severely threatened by devastating diseases, including Phytophthora blight (caused by oomycete *Phytophthora capsici*) and bacterial spot (e.g., *Xanthomonas spp.*), which can cause 20–30% yield losses annually. Breeding high-resistance varieties is fundamental for sustainable pepper production. However, the lack of a comprehensive NLR repertoire in pepper has impeded targeted breeding efforts. Among Solanaceae crops, NLR families in tomato (*Solanum lycopersicum*) [13] and potato (*Solanum tuberosum*) [14] have been systematically identified and analyzed, laying the foundation for understanding their disease resistance mechanisms. In contrast, pepper, with a later-completed genome sequence, has not been the subject of comprehensive NLR studies, hindering the utilization of modern molecular tools (e.g., gene editing, marker-assisted selection) to exploit its innate resistance resources, decipher pathogen interactions, and guide molecular breeding [15]. Many crops have identified genes associated with important diseases through genome-wide sequencing and analysis [16,17].

Therefore, this study leverages the high-quality pepper reference genome (‘Zhangshugang’) [18] to conduct the first genome-wide analysis of the NLR family in Capsicum annuum. We identified all potential NLR genes and systematically classified them based on domain architecture. We further dissected the contributions of gene duplication events (especially tandem and segmental duplication) to NLR family expansion and organization. Integrating transcriptome data from *P. capsici* infection and comparative genomic synteny analysis, we screened key candidate NLRs with potential disease resistance functions. Our findings elucidate the evolutionary dynamics of the pepper NLR family and its role in disease resistance, providing crucial insights for understanding pepper-specific immunity and revealing valuable candidate gene targets for the development of molecular markers for pepper resistance to *P*. *capsici*.

## 2. Materials and Methods

### 2.1. NLR Gene Identification

Protein sequences of *Arabidopsis* NLRs were retrieved from TAIR (https://www.arabidopsis.org/(accessed on 20 May 2025)). Homologous NLRs in pepper were identified using BLASTp against the pepper proteome (NCBI). HMMER v3.3.2 searched the whole proteome for core NLR domains (PF00931) using an E-value cutoff of 1 × 10^5^. Candidate sequences containing NB-ARC domains were retained, and redundancy was manually removed. Remaining candidates were validated via NCBI CDD (cd00204 for NB-ARC; https://www.ncbi.nlm.nih.gov/Structure/cdd/(accessed on 22 May 2025)) and Pfam batch search (https://www.ebi.ac.uk/interpro/(accessed on 22 May 2025)). N-terminal (TIR, CC, RPW8) and C-terminal (LRR) domains were checked for presence/completeness. Physicochemical parameters (aa length, MW, pI) were predicted using TBtools v2.360 Protein Parameter Calc.

### 2.2. Phylogenetic Analysis

NB-ARC domain (or full-length) sequences of pepper NLRs were aligned using Muscle v5 (--auto). Maximum Likelihood (ML) trees were constructed in IQ-TREE with 1000 bootstrap replicates. *Arabidopsis* and *Solanum lycopersicum* NLRs served as the outgroup.

### 2.3. Gene Duplication and Synteny Analysis

Synteny was analyzed using MCScanX in TBtools. Synteny plots were generated using Advanced Circos (TBtools v2.360). Orthologous NLRs between pepper and selected species were identified using Dual Synteny Plotter (TBtools v2.360).

### 2.4. Cis-Regulatory Element (CRE) Prediction

Promoter regions (2 kb upstream of TSS) were extracted. CREs were predicted using PlantCARE (https://bioinformatics.psb.ugent.be/webtools/plantcare/html/(accessed on 23 May 2025)), focusing on defense-related elements (SA/JA-responsive, WRKY-binding W-box).

### 2.5. RNA-Seq Analysis

Clean reads from *P. capsici*-infected CM334 (SRR9883231) and NMCA10399 (SRR9883230) were downloaded from NCBI (https://ngdc.cncb.ac.cn/gsa/browse/insdc/SRA1625870 assessed on 24 September 2025). Reads were mapped to the pepper genome using the Hisat2 plugin in TBtools. FPKM and differentially expressed genes were calculated with the DESeq2 plugin in TBtools, and the Benjamini–Hochberg method was employed to perform multiple hypothesis test correction on the hypothesis test probability (*p*-value), resulting in the error detection rate. The screening conditions for differentially expressed genes (DEGs) are |log2 Fold Change| ≥ 1 and FDR < 0.05. The differentially expressed genes were compared to the GO and KEGG databases to screen out the functional groups with *p* values less than 0.05. Correlation analysis was performed using SPSS Statistics 27. Heatmaps were generated with TBtools.

### 2.6. Protein–Protein Interaction (PPI) Network and Protein Structure Prediction

PPI networks were predicted using STRING(https://string-db.org/ (accessed on 8 June 2025); confidence >0.4). Protein structures were predicted using SWISS-MODEL (https://swissmodel.expasy.org/).

### 2.7. RT-qPCR Analysis

Plants were grown at the Variety Resource Garden of the Modern Agricultural Industrial Park (37°21′22.08′′ N, 116°20′4.71′′ E), affiliated with the Dezhou Academy of Agricultural Sciences, where the soil type is moist soil, with a primarily medium or light loam texture in the cultivated layer. The plants were sampled in July 2025, during which the average monthly temperature was 30.3 °C, the average relative humidity exceeded 70%, and the average precipitation for June to July was 134.2 mm, characterized by short-duration heavy rainfall events. The sampling time was 28 July 2025, with a temperature of 30 °C and a humidity of 93% on that day. The peppers were cultivated with a row spacing of 60 cm and plant spacing of 35–40 cm, employing drip irrigation with integrated water and fertilizer management. For transcriptome analysis, plants at the uniform mid-vegetative growth stage were selected, and sampling was performed when typical symptoms of phytophthora blight were observed.

Total RNA was extracted from the leaves, roots, and stems using TRIzol (3 biological replicates). cDNA was synthesized using TransScript^®^ One-Step gDNA Removal and cDNA Synthesis SuperMix. RT-qPCR was performed on a 7500 Real-Time PCR System with TB Green^®^ Premix Ex Taq^TM^ (Tli RNaseH Plus). Relative expression was calculated via 2^(−ΔΔCT)^ using *CaUBI-3* for normalization. The primers are listed in Appendix A. *CaUBI-3* primer efficiency was validated via a standard curve, with amplification specificity confirmed by melt curve analysis.

## 3. Results

### 3.1. Genome-Wide Identification and Classification of NLR Genes in Pepper

Using the ‘Zhangshugang’ genome and a stringent bioinformatics pipeline (based on conserved NLR domain architecture, NBS-LRR), we identified 288 high-confidence classical NLR genes. Classification based on N-terminal signaling domains revealed three major subfamilies: the predominant CC-NLRs (CNLs) accounted for 83% (238 genes), RPW8-NLRs (RNLs) constituted 9% (27 genes), and TIR-NLRs (TNLs) formed a smaller group at 8% (23 genes) (Figure 1); tomato and potato genomes were used for clustering. NLR protein lengths ranged from 487 aa (Caz01g38720) to 2270 aa (Caz09g02360), with a mean length of 1007 amino acids (aa). Molecular weights spanned 55,576.3 Da (Caz01g38720) to 260,534.8 Da (Caz09g02360), averaging 115,321.7 Da. Isoelectric points (pI) varied from 4.7 (Caz03g41900) to 9.5 (Caz07g14910), averaging 6.5 (Appendix A).

Chromosomal distribution analysis (Figure 2, Appendix A) revealed significant clustering. *CaNLR* genes were not evenly dispersed but concentrated in hotspots, notably on Chr09 (63 NLRs). Other clusters were observed on Chr01 (28 NLRs), Chr03 (28 NLRs), and Chr05 (36 NLRs). NLR clusters were predominantly localized to subtelomeric regions, consistent with rapid evolution in repeat-rich, recombination-prone regions.

**Figure 1 cimb-47-00867-f001:**
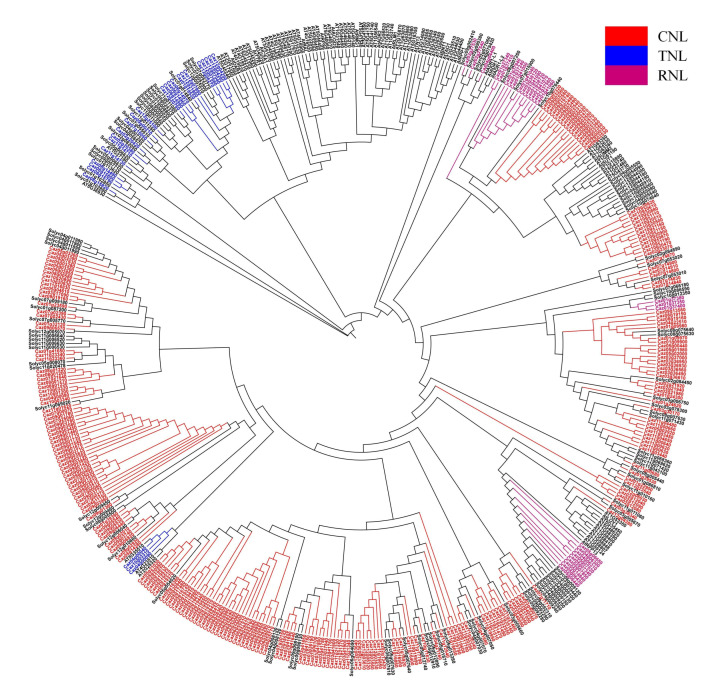
The phylogenetic evolutionary tree of Arabidopsis thaliana, tomato, and pepper was constructed using IQ-TREE with the maximum likelihood method (ML) and a step size of 1000. The gene IDs of NLRs from Arabidopsis and tomato are black. CaNLR genes are colored: red genes are TNL (TIR-NLR) type, blue genes are RNL (RPW8-NLR) type, purple genes are CNL (CC-NLR) type.

**Figure 2 cimb-47-00867-f002:**
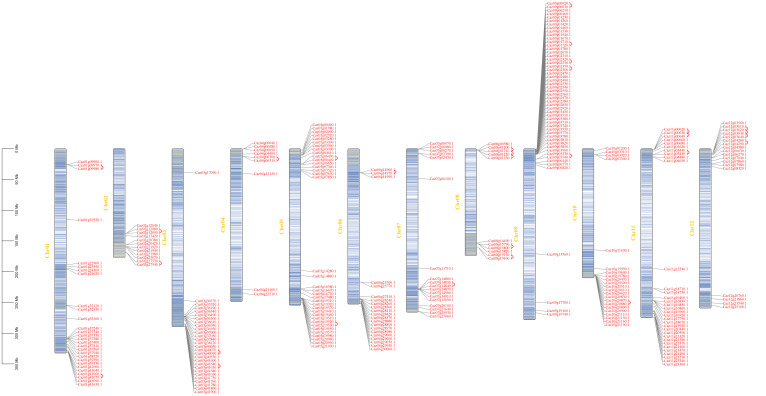
Physical map of 288 CaNLR genes on 12 chromosomes. The red arcs in the graph correspond to genes that have undergone tandem duplications.

### 3.2. Tandem Duplication Drives NLR Family Expansion

To elucidate the evolutionary mechanisms shaping the pepper NLR family, we analyzed the contributions of gene duplication events. Tandem duplication dominated, with 53 genes (18.4% of NLRs) identified across chromosomes (Figure 2 and Appendix A). Chr08 and Chr09 harbored ~55% of tandem duplicates. Segmental duplication accounted for only 5.5% of NLRs, with two segmental duplicates in Caz09g01530 and three in Caz07g00070 (Figure 3). These results highlight the major impact of local duplication on NLR repertoire expansion and diversification.

MCScanX analysis identified 68 orthologous NLR gene pairs between tomato (*Solanum lycopersicum*) and pepper (*Capsicum annuum*) (Figure 4, Appendix A), indicating high NLR homology between pepper and tomato.

### 3.3. NLR Expression Dynamics and Regulatory Element Analysis Reveal Immune-Related Genes

To predict the *CaNLR* roles in defense, we screened promoter regions (2 kb upstream of translation start sites) for *cis*-regulatory elements (CREs) (Appendix A). Figure 5 provides information on 46 key genes’ promoters. Bioinformatics analysis detected enrichment of defense-related hormone-responsive elements. Notably, 82.6% (238/288) of *CaNLR* promoters contained binding sites for salicylic acid (SA) and/or jasmonic acid (JA) signaling, including the JA/SA-responsive TGACG-box. Additionally, MYB motifs (TTGACC), recognized by immune-regulating MYB transcription factors, were identified in 171 *CaNLR* promoters, suggesting specific transcriptional regulation during defense activation.

The conserved motifs and domains were analyzed using MEME online software to determine the motif composition of NLR genes; a total of 10 conserved motifs were identified (Figure 6, Appendix A), named motifs 1–10. Motifs 1 and 6 repeatedly appear at the N-termini of the proteins and may determine their cellular localization. Motif 10 repeatedly appears at the C-termini of the proteins and may determine their functions in different positions, which is potentially related to the proteins’ key immunity function. Analysis of the CDSs and UTRs of NLR genes (Appendix A), specifically the 46 key genes shown in Figure 6, revealed that exon composition is generally conserved, with some genes, such as *Caz01g41040* and *Caz12g25100*, having particularly long intron sequences. The overall length of the gene sequence exceeds 12 kb, with introns accounting for the majority. This may be the result of a combination of evolutionary history and functional requirements (encoding, complex regulation), and there may be long noncoding RNA or intron regulatory sites present.

### 3.4. Disease Symptom Phenotyping in Resistant and Susceptible Cultivars and Organ-Specific Expression of NLR Candidates

To functionally validate *CaNLR*s’ involvement in pathogen defense, we analyzed expression profiles in resistant (CM334) and susceptible (NMCA10399) cultivars infected with *Phytophthora capsici* (Figure 7). Transcriptome data revealed 44 significantly differentially expressed NLR genes, with *Caz03g40070*, *Caz05g04610*, *Caz07g14830*, *Caz09g03770*, *Caz09g06440*, *Caz10g20900*, *Caz10g21150*, *Caz11g00620*, and *Caz11g00680* showing the most pronounced differences.

Furthermore, the expression levels of key resistance NLR genes were determined based on RT-qPCR of field-grown pepper materials with varying resistance. The results, together with the early phenotypes of stems, leaves, and fruits of disease-resistant and susceptible peppers grown in the field, are depicted in Figure 8. The stems of the susceptible material exhibited water-soaked, dark brown lesions that encircled the stem upwards, while the leaves showed water-soaked, dark green, irregular lesions with unclear edges. The fruits displayed water-soaked, dark green, blurry-edged spots. In contrast, the resistant material did not exhibit similar symptoms. Meanwhile, there are differences in the expression of key genes, with varying expression levels of different NLR genes across different materials and plant organs. *Caz09g03770* and *Caz10g21150* exhibit the highest expression levels in the stems of resistant materials, while *Caz03g40070* shows the highest expression levels in both the stems and leaves of resistant materials. *Caz10g20900*, on the other hand, has the highest expression level in the fruits of resistant materials. Upon sensing pathogen or stress signals, these genes are rapidly induced, ensuring the swift initiation of a potent immune response when pathogenic bacteria invade stem and leaf tissues. This effectively counteracts the invasion of Phytophthora from the stems or the infection of leaves through water droplet splash, effectively limiting the spread of disease lesions in the stems and leaves or rapidly triggering local defense responses in fruits, thereby supporting plant resistance.

### 3.5. Protein–Protein Interaction (PPI) Network and Protein Structure Prediction

STRING database analysis (medium confidence >0.4) predicted PPI networks involving NLRs (Figure 9). Among the 44 differentially expressed genes, 9 showed interactions, with Caz01g22900 and Caz09g03820 occupying central positions. Furthermore, the tertiary structures of proteins encoded by selected genes were modeled using SWISS-MODEL, and their subcellular localization was predicted. The results indicate that the 3D structural differences in encoded proteins may determine their disease resistance specificity, organ preference, and defense efficiency. The protein structures of Caz03g40070 and Caz09g03770 are similar, and both are located in the cytoplasm, with broad binding interfaces that can recognize multiple effector proteins. The N-terminal effector domains of Caz10g20900 and Caz10g21150 exhibit distinct helical bundle structures, which may mediate homologous oligomerization and directly recruit downstream signaling proteins. The C-terminal leucine-rich repeat domains of these proteins are composed of repeated β-folding–α-helix units that form a distinct horseshoe-shaped supercoiled structure, which may provide a structural basis for direct or indirect recognition of pathogenic effector proteins. In addition, the divergent helical bundles in N-terminal domains (e.g., *Caz10g20900*) compared with conserved *β*-*α* horseshoe folds in LRR domains imply distinct mechanisms for effector recognition and signal transduction.

## 4. Discussion

This study presents the genome-wide identification, evolutionary analysis, and functional prediction of the NLR family in pepper. We identified 288 high-confidence *CaNLR*s, demonstrated tandem duplication as the primary expansion driver, and pinpointed candidate genes responsive to pathogen stress and potential immune signaling hubs.

Tandem duplication was the predominant expansion mechanism (~18%), comparable to tomato (~21%) and potato (~18%) (Figure 2), suggesting a conserved Solanaceae strategy against distinct pathogen pressures. Pepper, facing diverse pathogens (viruses such as ToLCNDV, oomycetes such as *P. capsici*, bacteria such as *Xanthomonas* spp., and fungi such as *Leveillula taurica*), experiences strong NLR selection pressure [19,20]. Tandem duplication, particularly near telomeres, enables explosive local amplification of tightly linked gene copies. This genomic architecture favors ectopic recombination, accelerating the birth–death evolution of NLR alleles—a strategy potentially optimized against *P. capsici*’s rapid effector diversification. Such architecture accelerates new allele generation via unequal exchange and gene conversion, facilitating rapid adaptation to emerging pathogens [21]. This strategy underpins pepper’s resilience against diverse pathogens.

Subfamily composition showed significant bias: CC-NLRs (CNLs) predominated (80%), exceeding the proportion in *Arabidopsis* (~40%) but approaching that in tomato (~62%). This conserved CNL enrichment in Solanaceae may reflect their crucial role in recognizing effectors from Gram-negative bacteria (e.g., *Xanthomonas*-induced bacterial spot) [22]. Pepper’s susceptibility to *Xanthomonas* likely drives this CNL bias. Conversely, the few RNLs (27 genes) were all predicted to contain zinc-finger domains, reminiscent of *Arabidopsis* RPW8.1/8.2 proteins, which enhance callose deposition as a defense against fungal pathogens [23]. Thus, pepper RNLs may activate cell wall reinforcement defenses against fungi such as *L. taurica*. The conservation of the RPW8–zinc-finger combination suggests its functional importance in antifungal defense across Solanaceae. The scarcity of orthologous NLR pairs between pepper and *Arabidopsis* underscores lineage-specific innovation in Solanaceae NLR repertoires, possibly driven by host–pathogen coevolution within this clade.

Expression and co-expression analyses identified pathogen-responsive NLR candidates. *Caz09g03770* was strongly induced early during *P. capsici* infection (12-fold upregulation). Crucially, its tomato ortholog gene also responds to *Phytophthora*, indicating conserved roles in Solanaceae *Phytophthora* resistance. This gene is a prime target for homology-based cloning or cross-species editing. Conversely, species-specific clusters exist, such as the Chr03 tandem cluster, absent in tomato/potato, suggesting lineage-specific expansion against pepper pathogens (e.g., *L. taurica*). This coexistence of conserved and specialized NLRs reflects plants’ dual strategy: broad-spectrum conserved defenses combined with rapid local evolution for specific resistance. Using these results and combining multi-omics joint analysis methods and multiple technological means (including GWAS analysis or constructing genetic populations for QTL mapping) to develop molecular markers for pepper breeding is an important approach for future research and breeding of disease-resistant pepper varieties [24].

While this study maps the pepper NLR family and reveals key features, functional validation is imperative; future work should employ in planta assays (e.g., VIGS knockout) coupled with Y2H/Co-IP to verify effector recognition specificity. Furthermore, Caz01g22900.1 and Caz09g03820.1 were predicted as immune hubs, but their signaling mechanisms are unclear. Investigating whether pepper NLRs (e.g., *CaNLR42*) form signalosomes via liquid–liquid phase separation (LLPS) [25]—a key mechanism in animal innate immunity (e.g., MAVS, cGAS-STING) [26,27]—might be a potential future research direction. Beyond protein-level complex formation, future investigations should also extend to the post-transcriptional regulation of these identified *CaNLRs*. While our promoter analysis focused on transcriptional control, the stability, splicing, and translation efficiency of NLR mRNAs themselves are critical regulatory layers in mounting an effective immune response. Notably, mRNA modifications such as N6-methyladenosine (m6A) have emerged as key players in regulating plant stress responses [28]. It would be intriguing to explore whether the transcripts of the hub *CaNLRs* or those rapidly induced upon infection are subject to m6A modification, which could potentially fine-tune their expression dynamics and influence pepper’s disease outcomes. Integrating such epitranscriptomic analyses with our genomic findings will provide a more holistic understanding of NLR regulation in pepper immunity.

## 5. Conclusions

In summary, we present a tandem-duplication-driven expansion model, domain-specific functional implications, conserved/specialized candidate genes, and potential molecular marker targets. In the future, the value of these potential targets can be clarified through gene function verification, thereby providing new insights for research on the disease resistance of peppers.

## Figures and Tables

**Figure 3 cimb-47-00867-f003:**
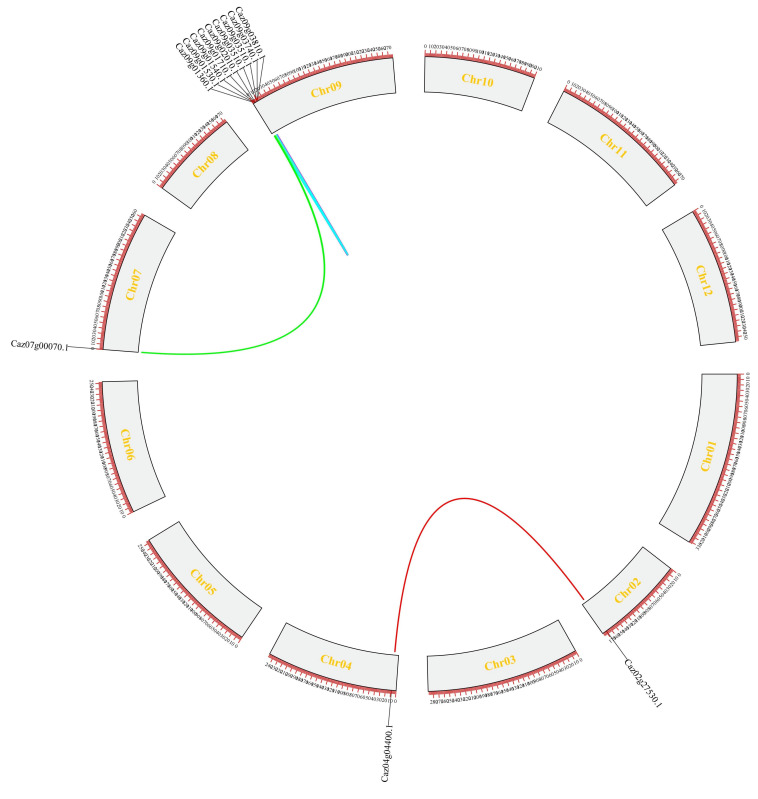
Segmental duplication of 288 CaNLR genes on 12 chromosomes. Genes linked with a different colors line are pairs of segmentally duplicated genes.

**Figure 4 cimb-47-00867-f004:**
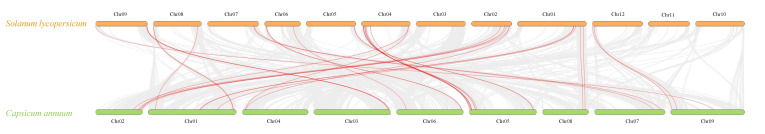
Syntenic relationships between homologous NLR genes of pepper and tomato. The NLR gene pairs between different species are highlighted with red lines.

**Figure 5 cimb-47-00867-f005:**
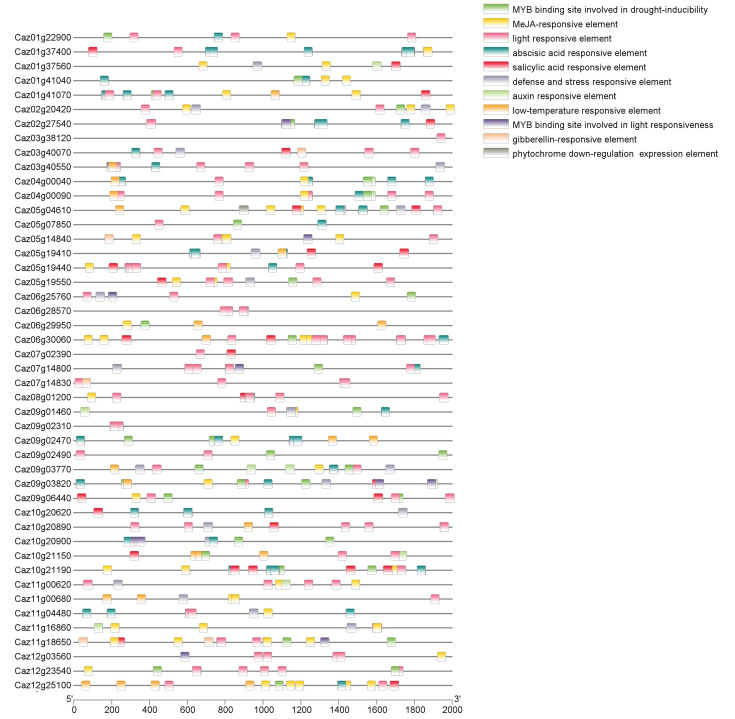
*Cis*-regulatory elements in the 2 kb upstream regions of 46 NLRs’ coding sequences. Rounded rectangles with different colors indicate different cis-acting elements.

**Figure 6 cimb-47-00867-f006:**
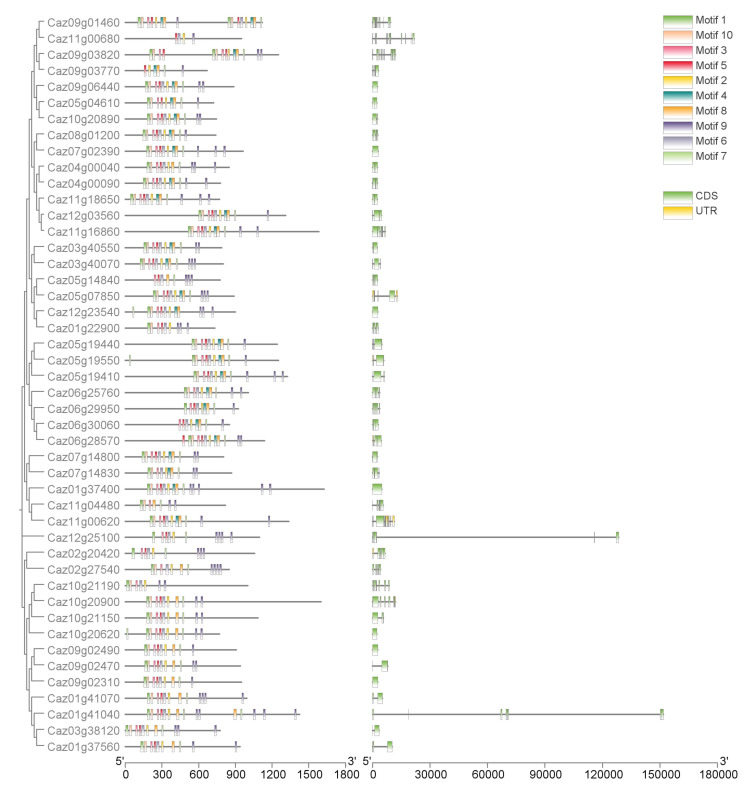
The phylogenetic relationships between NLR genes, constructed with bootstrap values of 1000 replicates. Conserved motifs and domains of 46 NLR genes were analyzed using MEME (**Left**). The gene structures of the NLR genes were analyzed using TBtools. CDSs and UTRs are colored with green and yellow boxes, respectively, where black lines represent introns (**Right**).

**Figure 7 cimb-47-00867-f007:**
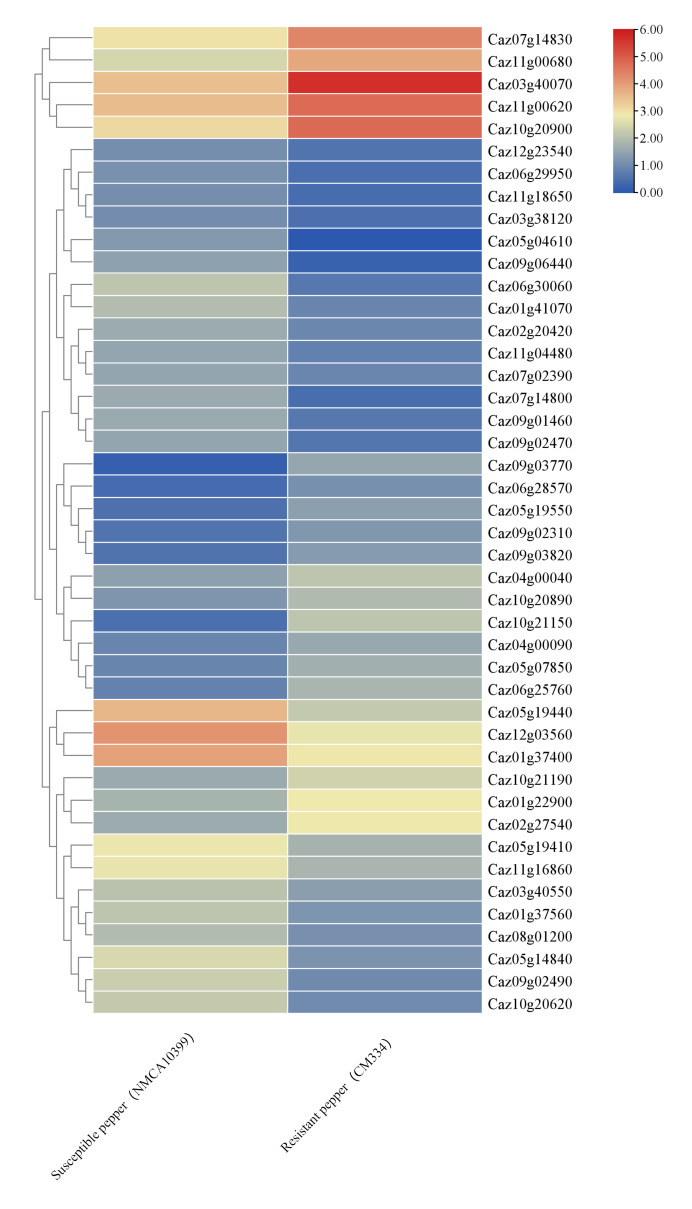
The expression levels of NLR genes in resistant and susceptible pepper cultivars infected with *Phytophthora capsici*. Redder colors indicate higher expression levels, and bluer colors indicate lower expression levels.

**Figure 8 cimb-47-00867-f008:**
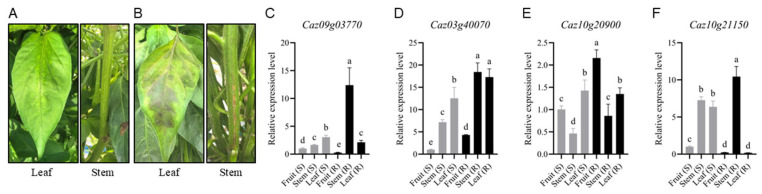
Early and mid-stage phenotypes of leaves and stems of resistant (**A**) and susceptible peppers (**B**) after infection with *Phytophthora capsici*. The expression analysis of *Caz09g03770* (**C**), *Caz03g40070* (**D**), *Caz10g20900* (**E**), and *Caz10g21150* (**F**) in resistant and susceptible pepper cultivars. The RT-qPCR results were analyzed using the 2^−(ΔΔCT)^ method. The data are shown as ± SE (*n* = 3). Different lowercase letters represent significant differences between different treatments (*p* < 0.05).

**Figure 9 cimb-47-00867-f009:**
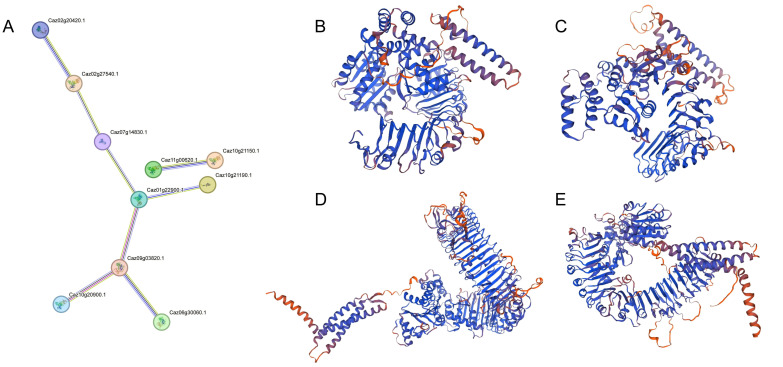
Protein–protein interaction network of NLR genes. Edges indicate protein–protein associations (**A**). Three-dimensional structure diagrams of important NLR proteins related to *Phytophthora capsici* resistance were simulated and generated using SWISS-MODEL software: Caz03g40070 (**B**), Caz09g03770 (**C**), Caz10g20900 (**D**), and Caz10g21150 (**E**).

## Data Availability

The original pepper genetic data presented in this study are publicly accessible in SRA930554 of the Genome Sequence Archive (GSA) (https://ngdc.cncb.ac.cn/gsa/browse/insdc/SRA930554 (accessed on 6 June 2025)). Other data are contained within the article or Appendix A.

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
