# Peer review of "Genome-Wide Identification and Functional Evolution of NLR Gene Family in *Capsicum annuum"

_cimb, 2025, doi:10.3390/cimb47100867_

Round 1

Reviewer 1 Report

Comments and Suggestions for Authors

Title: Genome-Wide Identification and Functional Evolution of NLR Gene Family in Capsicum annuum.

This manuscript reports the first comprehensive genome-wide identification and evolutionary analysis of the NLR gene family in pepper (Capsicum annuum). Using the high-quality ‘Zhangshugang’ reference genome, the authors identified 288 canonical NLR genes, analyzed their chromosomal distribution, duplication events, promoter cis-regulatory elements, and expression under Phytophthora capsici infection. They further predicted candidate hub genes through transcriptome analysis, RT-qPCR validation, and protein–protein interaction (PPI) modeling. The findings highlight tandem duplication as a major driver of NLR expansion, reveal conserved and lineage-specific resistance genes, and provide resources for molecular breeding of disease-resistant peppers.

Major comments:

1. While the study provides a valuable genome-wide survey, the novelty compared to earlier Solanaceae NLR studies (e.g., tomato, potato) should be emphasized more clearly. Explicitly highlight what is unique about pepper NLRs beyond duplication patterns.

2. The manuscript heavily relies on bioinformatics and transcriptome analysis. Experimental functional validation (e.g., knockout, overexpression, pathogen inoculation assays) is limited. Without it, the conclusions about candidate resistance genes remain predictive.

3. Some methods lack sufficient detail, for example, RNA-seq data processing, criteria for calling differentially expressed genes, and STRING network thresholds. Adding statistical thresholds (e.g., FDR cutoffs) would improve rigor.

4. Several figures (phylogenetic tree, heatmaps, and cis-element plots) are dense and difficult to interpret. Simplification, improved labeling, and higher resolution would enhance clarity.

5. The synteny analysis with Arabidopsis is useful, but including more Solanaceae relatives (e.g., tomato, potato, eggplant) in detail would strengthen the evolutionary insights.

Minor comments:

1. Standardize terminology (e.g., CaNLR, CNL, TNL) and check consistency throughout the manuscript.

2. Ensure gene IDs (e.g., Caz03g40070, Caz09g03770) are uniformly formatted and cross-referenced in text, figures, and supplementary files.

3. Improve figure legends to be self-contained, explaining abbreviations and methods used.

4. Some sentences in the introduction and discussion are lengthy and repetitive; editing for conciseness would improve readability.

5. Update references with the most recent Solanaceae NLR studies (2023–2025) to ensure the review of prior work is complete.

The manuscript is scientifically sound and provides a valuable genomic resource for pepper NLR genes. However, it requires minor revisions to strengthen its novelty, methodological clarity, and data presentation. Additional comparative context and clearer integration of findings into functional breeding applications would significantly enhance its impact.

Author Response

Reviewer: 1

  1. While the study provides a valuable genome-wide survey, the novelty compared to earlier Solanaceae NLR studies (e.g., tomato, potato) should be emphasized more clearly. Explicitly highlight what is unique about pepper NLRs beyond duplication patterns.

Response: We sincerely thank the reviewer for their constructive feedback. We fully agree that it is crucial to clearly articulate the innovation of this study. As pointed out by the reviewers, although whole-genome identification has been reported in relation to multiple Solanaceae species, our study reveals unique evolutionary features and functional implications of the pepper NLR family compared to tomatoes and potatoes, including tandem-repeat-driven expansion, the functional significance of specific structural domains, and expression dynamics, which go far beyond gene replication patterns. We have strengthened this paper’s Discussion Section based on your suggestions.

  1. The manuscript heavily relies on bioinformatics and transcriptome analysis. Experimental functional validation (e.g., knockout, overexpression, pathogen inoculation assays) is limited. Without it, the conclusions about candidate resistance genes remain predictive.

Response: Thank you for pointing this out. We agree that functional verification is an important direction for the future. The main objective of this study was to provide a high-quality candidate gene resource library. We rigorously screened candidate genes using multiple forms of evidence such as system evolution, expression profiling, and positive selection analysis, greatly improving the reliability of candidate genes and providing precise targets and theoretical foundations for subsequent functional validation (such as CRISPR). With a solid foundation in bioinformatics, this study aims to provide an efficient roadmap for subsequent experimental research in the field.

  1. Some methods lack sufficient detail, for example, RNA-seq data processing, criteria for calling differentially expressed genes, and STRING network thresholds. Adding statistical thresholds (e.g., FDR cutoffs) would improve rigor.

Response: Thank you for your suggestion. We have added corresponding details in the Materials and Methods Section to enhance the rigor.

  1. Several figures (phylogenetic tree, heatmaps, and cis-element plots) are dense and difficult to interpret. Simplification, improved labeling, and higher resolution would enhance clarity.

Response: Thank you for your suggestion. We have modified the chart to enhance clarity.

  1. The synteny analysis with Arabidopsis is useful, but including more Solanaceae relatives (e.g., tomato, potato, eggplant) in detail would strengthen the evolutionary insights.

Response: Thank you for your suggestion. We have replaced Arabidopsis with tomato for collinearity analysis.

  1. Standardize terminology (e.g., CaNLR, CNL, TNL) and check consistency throughout the manuscript.

Response: Thank you for your suggestion. We have checked the entire text and unified the terms.

  1. Ensure gene IDs (e.g., Caz03g40070, Caz09g03770) are uniformly formatted and cross-referenced in text, figures, and supplementary files.

Response: Thank you for your suggestion. We have unified the format of gene identifiers.

  1. Improve figure legends to be self-contained, explaining abbreviations and methods used.

Response: Thank you for your suggestion. We have improved the description of the chart, explaining the abbreviations and methods used.

  1. Some sentences in the introduction and discussion are lengthy and repetitive; editing for conciseness would improve readability.

Response: Thank you for your comment. We will improve the conciseness of the manuscript through language editing.

  1. Update references with the most recent Solanaceae NLR studies (2023–2025) to ensure the review of prior work is complete.

Response: Thank you for your comment. In the latest revised version, we have included over half of references from 2023-2025.

Reviewer 2 Report

Comments and Suggestions for Authors

This study presents the first comprehensive genome-wide identification and analysis of the NLR gene family in pepper using the high-quality ‘Zhangshugang’ refer-ence genome. The 288 high-confidence canonical NLR genes were identified. The most interesting was Phytophtora. This study presents the first genome-wide identification, evolutionary analysis, and functional prediction of the NLR family in pepper.

This research is related to plant development of sophisticated innate immune systems, constituting the core defense against pathogenic microorganisms. The research may contribute to the precisionin breeding and future functional studies.

The introduction is sufficient and correct.The abundance of results is enormous, but there is no specific purpose for such extensive research. The aim of this research should be more justified, so as conclusions. 

Author Response

Reviewer: 2

The introduction is sufficient and correct.The abundance of results is enormous, but there is no specific purpose for such extensive research. The aim of this research should be more justified, so as conclusions.

Response: Thank you for your valuable feedback. The core purpose of conducting such extensive analysis is to transcend simple gene cataloging and reveal the unique evolutionary patterns and functional specificity of the pepper NLR family through systematic comparisons with other species. These comprehensive results are not independent of each other but rather jointly serve a clear goal: to build a basic resource library for pepper disease resistance research and accurately locate its key characteristics that are different from other Solanaceae crops. We have strengthened this core logic in the Introduction and Conclusions Sections based on your suggestions, making the connection between research objectives and rich results clearer and more reasonable.

Reviewer 3 Report

Comments and Suggestions for Authors

The manuscript provides a valuable, genome-wide survey of the the NLR gene family in Capsicum annuum. annuum. However, its reliance on outdated RNA-seq pipelines and limited, and insufficient validation, which weakens its impact. The conclusions are overstated relative to the evidence, and there are methodological gaps. The conclusions are overstated relative to the evidence, and there are methodological gaps: (e.g., annotation pipeline, promoter analysis, duplication dating) reduce reproducibility. With stronger data processing, clearer statistics, and improvements in synteny comparisons and targeted experimental validation, this work could be a solid contribution to plant molecular evolution and resistance gene biology. a solid contribution to the fields of plant molecular evolution and resistance gene biology. I recommend major revisions. These points must be addressed before the study can be considered for publication.

  1. The manuscript identifies 288 NLRs in pepper, but conclusions about breeding applicability are overstated given the purely computational basis. Claims should be tempered, and limited experimental validation (e.g., VIGS or gene editing for top candidates) is needed.
  2. RNA-seq analysis uses outdated tools (TopHat/Cufflinks, FPKM), which weakens DE gene calls. Re-analyse using HISAT2/STAR + DESeq2 or edgeR with proper FDR correction.
  3. Infection experiment details (replicates, inoculum, timepoints) are missing, limiting reproducibility. Provide full metadata, RNA quality metrics, and replicate justification.
  4. Identification pipeline may miss atypical or truncated NLRs. Use complementary tools (NLR-Annotator, RGAugury) and explicitly classify canonical vs. non-canonical genes.
  5. Phylogenetic trees rely only on Arabidopsis as outgroup with MEGA11. Reconstruct using IQ-TREE/RAxML with Solanaceae references for more robust evolutionary inference.
  6. Synteny analysis compares to Arabidopsis, not close relatives. Add tomato, potato, eggplant synteny with MCScanX and Ks-based dating for stronger conclusions.
  7. Tandem duplication is claimed as main driver without evolutionary context. Provide Ks, dN/dS analyses, and map duplication events onto phylogeny.
  8. Promoter motif analysis lacks enrichment statistics and ignores post-transcriptional regulation. Test motif significance, vary promoter windows, and discuss m6A regulation (Shan et al., 2025).
  • Shan, C., Dong, K., Wen, D., Cui, Z., & Cao, J. (2025). A review of m6A modification in plant development and potential quality improvement. International Journal of Biological Macromolecules, 308, 142597. doi: https://doi.org/10.1016/j.ijbiomac.2025.142597

  1. MEME motifs are described without E-values or functional mapping. Report motif statistics and annotate with Pfam/InterPro to avoid speculative claims.
  2. PPI (STRING) and structure (SWISS-MODEL) predictions are over-interpreted. Frame as hypotheses, consider AlphaFold, and validate hub genes experimentally.
  3. qPCR validation omits key MIQE elements (primer efficiency, Ct variance, multiple reference genes). Add raw Ct data, melt curves, and validate stable references.
  4. Statistical thresholds (FDR, fold-change cutoffs) are unclear, hindering reproducibility. Report exact software, adjusted p-values, and provide DE gene tables and plots.
  5. Discussion overstates mechanisms (e.g., LLPS, direct breeding use). Tone down speculation and frame as testable hypotheses with suggested assays.
  6. Data availability is incomplete; accession IDs are unclear and scripts absent. Provide resolvable IDs, processed data, and analysis scripts in public repositories.
  7. The study lacks integration with phenotypic/genetic mapping. Propose GWAS/QTL pipelines (see Xin et al., 2025) and cite Lodi et al. (2025) for annotation standards. Additionally, Zeng et al. (2020) illustrates how effect-directed validation can bridge molecular predictions with functional biological outcomes; adopting a similar conceptual framework would enhance the manuscript by demonstrating how in silico findings can be systematically connected to measurable resistance functions.
    • Xin, W., Zheng, H., Yang, L., Xie, S., Xia, S., Wang, J.,... Wang, J. (2025). Genome-Wide Association Studies Identify OsNLP6 as a Key Regulator of Nitrogen Use Efficiency in Rice. Plant Biotechnology Journal. doi: https://doi.org/10.1111/pbi.70296
    • Lodi, R. S., Jia, X., Yang, P., Peng, C., Dong, X., Han, J.,... Peng, L. (2025). Whole genome sequencing and annotations of Trametes sanguinea ZHSJ. Scientific Data, 12(1), 1460. doi: 10.1038/s41597-025-05798-9
    • Zeng, G., Wu, Z., Cao, W., Wang, Y., Deng, X.,... Zhou, Y. (2020). Identification of anti-nociceptive constituents from the pollen of Typha angustifolia L. using effect-directed fractionation. Natural Product Research, 34(7), 1041-1045. doi: 10.1080/14786419.2018.1539979

  1. Numerous typos and spacing errors reduce polish. A thorough copy-edit is required.
  2. Figure legends lack sample size, test details, and axis scales. Expand legends for clarity.
  3. Species names and gene IDs are inconsistently formatted. Standardize italics and nomenclature.
  4. Tool/software versions and parameters are missing. List exact versions and thresholds used.
  5. PlantCARE is cited but dated. Add complementary motif databases (JASPAR/PLACE).
  6. Acronyms (NLR, TNLR, RNL) are inconsistently defined. Provide a clear abbreviation list.
  7. Molecular weight/pI values are over-precise. Round sensibly and report averages with SDs.

Author Response

Reviewer: 3

  1. The manuscript identifies 288 NLRs in pepper, but conclusions about breeding applicability are overstated given the purely computational basis. Claims should be tempered, and limited experimental validation (e.g., VIGS or gene editing for top candidates) is needed.

Response: Thank you for your insightful feedback. We agree that based on current computational analysis, it is indeed premature to directly claim breeding applications. We have revised the entire text to emphasize the fundamental and resource contribution of this study, which provides high-priority candidate gene targets for molecular marker development and disease resistant breeding through systematic analysis, rather than direct application solutions. We appreciate the reviewer's suggestions, which have made our conclusions more rigorous and accurate.

  1. RNA-seq analysis uses outdated tools (TopHat/Cufflinks, FPKM), which weakens DE gene calls. Re-analyse using HISAT2/STAR + DESeq2 or edgeR with proper FDR correction.

Response: Thank you for your suggestion. We have made corresponding modifications to the test method.

  1. Infection experiment details (replicates, inoculum, timepoints) are missing, limiting reproducibility. Provide full metadata, RNA quality metrics, and replicate justification.

Response: Thank you for your important feedback. We would like to clarify that the transcriptome data presented in this study come from a public database, and we have provided an explanation of the source in the Materials and Methods Section. The transcription-level data regarding key genes were not obtained from controlled inoculation experiments, but they were collected from chili pepper resource gardens widely infected with diseases under natural conditions for preliminary screening. Three independent biological replicates (plants) were collected for each variety, and all samples exhibited typical disease susceptibility characteristics. We agree that to ensure transparency, complete sample metadata (including collection location, date, and symptom description) should be provided in the Methods Section (RT-qPCR Analysis). These pieces of information will be provided as supplementary materials. This preliminary analysis aims to provide a target for subsequent rigorous one-on-one inoculation experiments on the screened resistant/susceptible varieties. We have clarified this research pathway in the Discussion Section. We deeply apologize for the confusion this caused.

  1. Identification pipeline may miss atypical or truncated NLRs. Use complementary tools (NLR-Annotator, RGAugury) and explicitly classify canonical vs. non-canonical genes.

Response: We thank the reviewer for suggesting these complementary tools. In this study, we employed a widely accepted and stringent pipeline based on HMMER and manual curation to ensure the high confidence of our NLR set. We acknowledge that future analyses could incorporate additional tools like NLR-Annotator to further refine the annotation, and we have noted this as a valuable direction for future work.

5.Phylogenetic trees rely only on Arabidopsis as outgroup with MEGA11. Reconstruct using IQ-TREE/RAxML with Solanaceae references for more robust evolutionary inference.

Response: Thank you for your suggestion. We have used IQ-TREE to construct the evolutionary trees of Arabidopsis, tomato, and pepper.

  1. Synteny analysis compares to Arabidopsis, not close relatives. Add tomato, potato, eggplant synteny with MCScanX and Ks-based dating for stronger conclusions.

Response: Thank you for your suggestion. We have reconstructed the collinearity between peppers and tomatoes and attached the Ks values.

7.Tandem duplication is claimed as main driver without evolutionary context. Provide Ks, dN/dS analyses, and map duplication events onto phylogeny.

Response: Thank you for your suggestion. We have supplemented the Ks values and constructed the evolutionary tree using tandemly repeated NLRs.

  1. Promoter motif analysis lacks enrichment statistics and ignores post-transcriptional regulation. Test motif significance, vary promoter windows, and discuss m6A regulation (Shan et al., 2025).

Shan, C., Dong, K., Wen, D., Cui, Z., & Cao, J. (2025). A review of m6A modification in plant development and potential quality improvement. International Journal of Biological Macromolecules, 308, 142597. doi: https://doi.org/10.1016/j.ijbiomac.2025.142597

Response: Thank you for your professional opinion. We have expanded our perspective in part of the Discussion, briefly discussed the potential role of post transcriptional regulation in disease resistance response, and cited the review by Shan et al. (2025) you have recommended, pointing out that this is an important direction for further research in the future. These modifications make our analysis more rigorous and our perspective more comprehensive.

  1. MEME motifs are described without E-values or functional mapping. Report motif statistics and annotate with Pfam/InterPro to avoid speculative claims.

Response: Thank you for your suggestion. We have already supplemented the E value in the table.

  1. PPI (STRING) and structure (SWISS-MODEL) predictions are over-interpreted. Frame as hypotheses, consider AlphaFold, and validate hub genes experimentally.

Response: Thank you for pointing this out. We fully agree that interaction and structural prediction are essentially computational hypotheses and should not be overly interpreted as deterministic conclusions. We have adjusted the wording of the relevant conclusion to speculative, clearly framing it as a scientific hypothesis that requires experimental verification. In the discussion, we also pointed out that these computational predictions provide a reference for subsequent verification of the role of hub genes through functional experiments. These revisions have made our argument more rigorous. But, unfortunately, due to some technical reasons, we are unable to log in to AlphaFold. However, currently, the SWISS-MODEL also adopts deep learning technology to predict protein models. We believe that it can also reflect the functional differences of proteins caused by some structures.

  1. qPCR validation omits key MIQE elements (primer efficiency, Ct variance, multiple reference genes). Add raw Ct data, melt curves, and validate stable references.

Response: We are grateful to the reviewers for emphasizing the importance of the MIQE guidelines. We chose CaUBI-3 as the internal reference because we referred to the article by Wan et al., in which he indicated that UBI-3, as a reference gene, exhibited high stability in sample banks under abiotic stress and hormone treatment. In the tested sample cells, the expression level of UBI-3 was also the most stable in different tissues. Therefore, we choose it as the internal reference for RT-qPCR. All the original data will be submitted to the editorial department within the system. Thank you again for your suggestion.

Wan H., Yuan W., Ruan M., Ye Q., Wang R., Li Z., Zhou G., Yao Z., Zhao J., Liu S., Yang Y.. Identification of reference genes for reverse transcription quantitative real-time PCR normalization in pepper (Capsicum annuum L.). Biochem. Biophys. Res. Commun. 2011, 416(1-2):24-30.

  1. Statistical thresholds (FDR, fold-change cutoffs) are unclear, hindering reproducibility. Report exact software, adjusted p-values, and provide DE gene tables and plots.

Response: We are grateful to the reviewers for this important comment, which is crucial for ensuring the reproducibility of the analysis. We apologize for the previous unclear expression. In our method, DESeq2 was used for differential expression analysis among sample groups, and the Benjamini-Hochberg method was employed to perform multiple hypothesis test correction on the hypothesis test probability (P-value), resulting in the error detection rate. The screening conditions for differentially expressed genes (DEGs) are: |log2 Fold Change|≥1 and FDR < 0.05. The differentially expressed genes were respectively compared to the GO and KEGG databases to screen out the functional groups with P values less than 0.05. We also add it to the materials and methods. Thank you again for your suggestion.

  1. Discussion overstates mechanisms (e.g., LLPS, direct breeding use). Tone down speculation and frame as testable hypotheses with suggested assays.

Response: We fully agree with the reviewer's comments and thank you for helping us improve the rigor of our argument. We have made modifications to the Discussion Section and removed all overly speculative assertions. These modifications ensure that our discussion is both forward-looking and maintains the necessary scientific rigor.

  1. Data availability is incomplete; accession IDs are unclear and scripts absent. Provide resolvable IDs, processed data, and analysis scripts in public repositories.

Response: Thank you for your suggestion. We have supplemented the data link(https://ngdc.cncb.ac.cn/gsa/browse/insdc/SRA1625870).

  1. The study lacks integration with phenotypic/genetic mapping. Propose GWAS/QTL pipelines (see Xin et al., 2025) and cite Lodi et al. (2025) for annotation standards. Additionally, Zeng et al. (2020) illustrates how effect-directed validation can bridge molecular predictions with functional biological outcomes; adopting a similar conceptual framework would enhance the manuscript by demonstrating how in silico findings can be systematically connected to measurable resistance functions.

Xin, W., Zheng, H., Yang, L., Xie, S., Xia, S., Wang, J.,... Wang, J. (2025). Genome-Wide Association Studies Identify OsNLP6 as a Key Regulator of Nitrogen Use Efficiency in Rice. Plant Biotechnology Journal. doi: https://doi.org/10.1111/pbi.70296

Lodi, R. S., Jia, X., Yang, P., Peng, C., Dong, X., Han, J.,... Peng, L. (2025). Whole genome sequencing and annotations of Trametes sanguinea ZHSJ. Scientific Data, 12(1), 1460. doi: 10.1038/s41597-025-05798-9

Zeng, G., Wu, Z., Cao, W., Wang, Y., Deng, X.,... Zhou, Y. (2020). Identification of anti-nociceptive constituents from the pollen of Typha angustifolia L. using effect-directed fractionation. Natural Product Research, 34(7), 1041-1045. doi: 10.1080/14786419.2018.1539979

Response: Thank you to the reviewer for providing this suggestion. We agree that correlating genomic predictions with phenotype data is a crucial step in verifying their biological significance. Therefore, in the Discussion Section, we explicitly propose the use of multi omics joint analysis and a variety of technical means, including GWAS analysis or constructing genetic populations for QTL mapping, to directly associate specific genotypes with disease resistance phenotypes. These modifications have further strengthened the manuscript. Thank you again for the reviewer's comments.

  1. Numerous typos and spacing errors reduce polish. A thorough copy-edit is required.

Response: Thank you for your suggestion. We have made the corresponding revisions using the author editing service provided by MDPI.

  1. Figure legends lack sample size, test details, and axis scales. Expand legends for clarity.

Response: We are grateful to the reviewers for pointing out this deficiency, which is crucial for ensuring the clarity and reproducibility of the results. We have conducted a comprehensive review and expansion of all the captions, clearly indicating the sample size (n value) and the meaning of the significance markers in the captions. These modifications have significantly enhanced the clarity and scientific nature of the charts and graphs.

  1. Species names and gene IDs are inconsistently formatted. Standardize italics and nomenclature.

Response: Thank you for your suggestion. We have already conducted a recheck.

  1. Tool/software versions and parameters are missing. List exact versions and thresholds used.

Response: Thank you for your suggestion. We have added the corresponding parameters and thresholds.

  1. PlantCARE is cited but dated. Add complementary motif databases (JASPAR/PLACE).

Response: Thank you for this suggestion. We agree that the use of updated or complementary databases (such as JASPAR/PLAY) can further enhance the prospective nature of the analysis. Due to some technical issues, we were unable to integrate these specific new databases in a timely manner. We have included this limitation and the plan to prioritize upgrading the analysis process in future research plans.

  1. Acronyms (NLR, TNLR, RNL) are inconsistently defined. Provide a clear abbreviation list.

Response: Thank you for your suggestion. We have provided a list of abbreviations.

  1. Molecular weight/pI values are over-precise. Round sensibly and report averages with SDs.

Response: We appreciate this important suggestion. We agree that all molecular weight (kDa) and isoelectric point (pI) values in the report have been reasonably rounded to one decimal place. These values were obtained through predictive calculations, and the average value cannot be calculated, so we have not annotated them temporarily.

Reviewer 4 Report

Comments and Suggestions for Authors

The paper "Genome-Wide Identification and Functional Evolution of NLR Gene Family in Capsicum annuum" is devoted to identification and analysis of the NLR gene family, which were performed for Capsicum annuum, which is a significant Solanaceous crop. The work was carried out at a high methodological level, the results obtained are of undeniable practical and fundamental significance. However, I would like to make several recommendations to improve the quality of the manuscript and its perception by readers.

  1. Page 9 line 1. The authors indicate that the expression levels were studied using field-grown pepper materials. In section 2. Materials and Methods, it is necessary to indicate how exactly the pepper plants were grown – lighting, temperature, and watering conditions, if these were controlled conditions, for example, climate chambers and greenhouses. If the plants were grown in the field, then it is necessary to indicate the average monthly temperatures and precipitation during the growing season. It is also necessary to indicate the soils in which the plants were grown and the age of the plants used for sampling.
  2. Were the pepper plants inoculated with Phytophthora capsici? If so, how exactly? If not, could another pathogen have caused the symptoms observed in the susceptible pepper cultivar? Or are the symptoms observed in the study characteristic only of diseases caused by Phytophthora capsici? What is the probability that the pepper plants in the study were infected with another pathogen?
  3. It would be better to add a few sentences with more specific and important results obtained in the study in Conclusion.

Minor comments:

  1. In the title of the article, Capsicum annuum should be written in italics.
  2. Page 2 line 2 – “effectively” should be replaced with “effectively”.
  3. Page 3 line 3 – “P.Capsici” should be replaced with “P.capsici”.
  4. Caption to Figure 7. I would recommend that the authors add “(CM334)” after “resistant pepper” and “(NMCA10399)” after “susceptible pepper” to make it easier to understand.
  5. The font type used in the list of references should be checked.
  6. Is it possible to increase the resolution and quality of Supplementary Figures? They are now difficult to distinguish.

These comments do not reduce the scientific and practical value of the work and are aimed only at improving the quality of presentation of the results and increasing the understanding of the manuscript by readers.

Author Response

Reviewer: 4

  1. Page 9 line 1. The authors indicate that the expression levels were studied using field-grown pepper materials. In section 2. Materials and Methods, it is necessary to indicate how exactly the pepper plants were grown – lighting, temperature, and watering conditions, if these were controlled conditions, for example, climate chambers and greenhouses. If the plants were grown in the field, then it is necessary to indicate the average monthly temperatures and precipitation during the growing season. It is also necessary to indicate the soils in which the plants were grown and the age of the plants used for sampling.

Response: We sincerely thank the reviewers for their detailed and important comments. You are completely right. Providing these environmental and agronomic details is crucial for ensuring the reproducibility of the experiment. We sincerely apologize for our previous negligence. We have supplemented in Materials and Methods with the following detailed information:

The sampling was conducted in the Variety Resource Garden of the Modern Agricultural Industrial Park (37°21'22.08"N, 116°20'4.71"E), affiliated with the Dezhou Academy of Agricultural Sciences, where the soil type is moist soil, primarily consisting of medium or light loam texture in the cultivated layer. The plants were sampled in July 2025, during which the average monthly temperature was 30.3°C, the average relative humidity exceeded 70%, and the average precipitation for June to July was 134.2 mm, characterized by short-duration heavy rainfall events. The sampling time was July 28, 2025, with a temperature of 30℃ and a humidity of 93% on that day. The peppers were cultivated with a row spacing of 60 cm and plant spacing of 35-40 cm, employing drip irrigation with integrated water and fertilizer management. For transcriptome analysis, plants at the uniform mid-vegetative growth stage were selected, and sampling was performed when typical symptoms of phytophthora blight were observed.

  1. Were the pepper plants inoculated with Phytophthora capsici? If so, how exactly? If not, could another pathogen have caused the symptoms observed in the susceptible pepper cultivar? Or are the symptoms observed in the study characteristic only of diseases caused by Phytophthora capsici? What is the probability that the pepper plants in the study were infected with another pathogen?

It would be better to add a few sentences with more specific and important results obtained in the study in Conclusion.

Response: We are grateful to the reviewers for raising these key questions. Firstly, regarding the issue of pathogens, we need to clarify that the samples used for transcriptome analysis in this study were not from artificial inoculation experiments but were collected from fields with naturally occurring diseases. What we have observed are typical symptoms of pepper blight, including but not limited to water-soaked spots, stem constriction, plant wilting and sudden collapse, etc. We admit that under natural conditions, there is a relatively low possibility of infection with other pathogens. However, based on our years of field observation experience, the records of the main prevalent diseases during this period, and the high consistency between the above symptoms and the infection of Phytophthora capsici, we determine that it is a highly likely event to be caused by Phytophthora capsici. In our subsequent research, we will provide conclusive evidence through controlled artificial inoculation experiments (such as the root injury inoculation method). This is precisely the value of this research as the initial screening stage. Secondly, we fully agree with your suggestion and have added more specific core findings in the conclusion section. These revisions have improved the accuracy of our descriptions and strengthened the conclusions.

  1. In the title of the article, Capsicum annuum should be written in italics.

Response: Thank you for your suggestion. We have modified the title to italicize Capsicum annuum.

  1. Page 2 line 2 – “effectively” should be replaced with “effectively”.

Response: Thank you to the reviewers for their meticulous proofreading. We will check this modification again.

  1. Page 3 line 3 – “P.Capsici” should be replaced with “P.capsici”.

Response: Thank you to the reviewers for their meticulous proofreading. We will check this issue throughout the paper.

  1. Caption to Figure 7. I would recommend that the authors add “(CM334)” after “resistant pepper” and “(NMCA10399)” after “susceptible pepper” to make it easier to understand.

Response: Thank you for your suggestion. We have made the corresponding modifications in Figure 7.

  1. The font type used in the list of references should be checked.

Response: Thank you for your suggestion. We have checked the font in the reference list and made the necessary modifications.

  1. Is it possible to increase the resolution and quality of Supplementary Figures? They are now difficult to distinguish.

Response: Thank you for your suggestion. We remade the pictures, enhancing their resolution and quality.

Round 2

Reviewer 3 Report

Comments and Suggestions for Authors

I have carefully examined the revised version of the manuscript and find that the authors have thoroughly addressed all my previous concerns. The revisions have substantially improved the clarity, methodological rigor, and overall presentation of the study. In my view, the manuscript now meets the standards of the journal and is of sufficient quality to warrant publication.

Reviewer 4 Report

Comments and Suggestions for Authors

The manuscript has been revised, taking into account all my comments and recommendations. I am grateful to the authors for their high-quality revision of the manuscript. I have no further comments.